# MTSNet: Joint Feature Adaptation and Enhancement for Text-Guided Multi-view Martian Terrain Segmentation

Yang Fang
Chongqing University of Posts and
Telecommunications
Chongqing, China
fangyang@cqupt.edu.cn

Xuefeng Rao
Chongqing University of Posts and
Telecommunications
Chongqing, China
s220231077@stu.cqupt.edu.cn

Xinbo Gao*
Chongqing University of Posts and
Telecommunications
Chongqing, China
gaoxb@cqupt.edu.cn

Weisheng Li
Chongqing University of Posts and
Telecommunications
Chongqing, China
liws@cqupt.edu.cn

Zijian Min
Inha University
Incheon, South Korea
minzijian@inha.edu

## ABSTRACT

Martian terrain segmentation plays a crucial role in autonomous navigation and safe driving of Mars rovers as well as global analysis of Martian geological landforms. However, most deep learning-based segmentation models cannot effectively handle the challenges of highly unstructured and unbalanced terrain distribution on the Martian surface, thus leading to inadequate adaptability and generalization ability. In this paper, we propose a novel multi-view Martian Terrain Segmentation framework (MTSNet) by developing an efficient Martian Terrain text-Guided Segment Anything Model (MTG-SAM) and combining it with a tailored Local Terrain Feature Enhancement Network (LTEN) to capture intricate terrain details. Specifically, the proposed MTG-SAM is equipped with a Terrain Context attention Adapter Module (TCAM) to efficiently and effectively unleashing the model adaptability and transferability on Mars-specific terrain distribution. Then, a Local Terrain Feature Enhancement Network (LTEN) is designated to compensate for the limitations of MTG-SAM in capturing the fine-grained local terrain features of Mars surface. Afterwards, a simple yet efficient Gated Fusion Module (GFM) is introduced to dynamically merge the global contextual features from MTG-SAM encoder and the local refined features from LTEN module for comprehensive terrain feature learning. Moreover, the proposed MTSNet enables terrain-specific text as prompts resolving the efficiency issue of existing methods that require costly annotation of bounding boxes or foreground points. Experimental results on AI4Mars and ConeQuest datasets demonstrate that our proposed MTSNet can effectively learns the unique Martian terrain feature distribution and achieves state-of-the-art performance on multi-view terrain segmentation from both

the perspectives of the Mars rover and the satellite remote sensing. Code is available at https://github.com/raoxuefeng/mtsnet.

## CCS CONCEPTS

• **Computing methodologies → Image segmentation**.

## KEYWORDS

Martian Terrain Segmentation, Feature Enhancement, Contextual Attention Adapter, Gated Fusion, Text Prompt Encoder

**ACM Reference Format:**
Yang Fang, Xuefeng Rao, Xinbo Gao, Weisheng Li, and Zijian Min. 2024. MTSNet: Joint Feature Adaptation and Enhancement for Text-Guided Multi-view Martian Terrain Segmentation. In *Proceedings of the 32nd ACM International Conference on Multimedia (MM'24), October 28-November 1, 2024, Melbourne, Australia.* ACM, New York, NY, USA, 10 pages. https://doi.org/10.1145/3664647.3681430

## 1 INTRODUCTION

Since the 1960s, humans have conducted multiple Mars exploration missions. In recent years, with the advancement of deep space exploration technology, Mars exploration has once again become a popular topic. As relevant exploration missions progress, the increasing Mars data has greatly promoted the application of deep learning in Mars exploration [3, 12, 31, 33]. In Mars exploration, Martian terrain semantic segmentation plays a crucial role as the foundation for autonomous navigation and safe driving of Mars rovers [13], and can also assist satellites in precisely analyzing the global geological landforms of Mars, providing support and decision-making basis for Mars exploration programs.

Currently, semantic segmentation technology has made significant progress in fields such as autonomous driving [4, 7], human-computer interaction [37], and medical image analysis [11, 20]. However, these methods will encounter unprecedented challenges when applied to the extremely unstructured environments like the Martian surface [30]. The characteristics of Martian landforms are highly unstructured, lacking obvious structural features with regular geometric shapes or texture patterns on the Martian surface [10]. These factors make it infeasible to directly apply existing semantic segmentation models trained on common datasets to the task of learning Martian terrain features. Moreover, the images collected

---
*Corresponding author (gaoxb@cqupt.edu.cn).

by Mars rover and orbiting satellite [27] at different locations and perspectives have distinct shapes and large differences in imaging scale, further increasing the difficulty of accurate terrain semantic segmentation with current semantic segmentation methods.

In recent years, the emergence of foundation vision models has attracted widespread attention [5, 45]. These models, based on deep learning architectures and trained on large-scale and diverse image datasets, have acquired powerful feature extraction and zero-shot generalization capabilities. This means that even for unknown scenarios, after appropriate adjustments or transfer learning, they can still adapt to the semantic knowledge of new scenarios [38]. The recently emerging Segment Anything Model (SAM) [19], as a foundation model for prompt-based segmentation tasks, can segment objects of interest given a semantic prompt in the form of bounding boxes, points, or masks [38]. Several recent works have proposed strategies to reduce the computation costs of SAM. FastSAM [42] develops a CNN-based architecture to segment all objects in an image for efficiency improvement. MobileSAM [44] presents a decoupled distillation for obtaining a lightweight image encoder of SAM. EfficientSAM [48] leverages masked image pretraining, which learns to reconstruct features from SAM image encoder for effective visual representation learning. Although SAM has achieved excellent segmentation performance in general scenarios, due to the significant difference in data distribution between the source domain and the target domain, its performance still falls into a local optimum when applied to specific distribution data of the Martian surface environment [2, 5]. Besides, SAM and most of its variants are usually only capable of using prompts in the form of points, bounding boxes, etc., for semantic segmentation, overlooking the great potential of text prompts, which can greatly reduce annotation costs and speed up the training process.

To address the above issues, we propose MTSNet framework shown in Figure 1. While retaining the SAM model and inspired by various Adapter approaches [8, 16, 35], we present a novel Terrain Context Attention Adapter Module (TCAM) to enhance the model's adaptability and generalization capability for Martian images through fine-tuning strategy. Meanwhile, we construct a Local Terrain Feature Enhancement Network (LTEN), which captures local details in Martian terrain through stacks of efficient convolution operators and multi-scale attentions, compensating for the global contextual terrain features from MTG-SAM encoder and thus generating refiner segmentation results. To better fuse the contextual features from MTG-SAM and the local features from LTEN, we design a Gated Fusion Module (GFM) to dynamically fuse them through a gating mechanism to realize adaptive and learnable feature fusion process. Finally, we introduce a text prompt encoder enabling the model to perform efficient terrain segmentation by simple terrain text prompts, such as soil, sand, and etc., resolving the difficulty of using bounding boxes or foreground points as prompts in the complex and unstructured Martian environment. In summary, our contributions are as follows:

(1) To the best of our knowledge, our proposed MTSNet is the first to present and address the multi-view Martian Terrain Segmentation task with only text prompts and report superior performance on two representative benchmarks with significant flexibility and efficiency.

(2) We propose an efficient MTG-SAM encoder equipped with a lightweight Terrain Contextual Attention Adapter Module (TCAM), which is able to fine-tune the SAM model with extremely few adapter parameters, allowing it to quickly and adequately adapt to the target domain data.

(3) We design a lightweight terrain fine-grained feature learning network with stacks of convolution operators and multi-scale attention blocks, which can well compensate for the limitation of the MTG-SAM encoder in terrain detail capture.

(4) For better feature fusion, we designed a simple yet efficient gated fusion module (GFM) to dynamically fuse contextual and fine-grained features through an adaptive and learnable gating mechanism, which exhibits superior effectiveness.

## 2 RELATED WORK

### 2.1 Martian Terrain Segmentation

Martian terrain segmentation is a special and important subset of semantic segmentation, with the goal of segmenting different terrains from diverse Martian images. Whether it is the analysis of Martian landforms and terrain by satellites or tasks such as autonomous obstacle avoidance, path planning, and terrain traversability judgment for Mars rovers [13], all are profoundly influenced by Martian terrain segmentation. In the research of Martian terrain segmentation, Liu et al. [24] proposed a Hybrid Attention Semantic Segmentation (HASS) network that combines global intra-class and local inter-class pixel attention mechanisms. Swan et al. [40] collected a Martian terrain segmentation dataset called AI4Mars and evaluated its performance using DeepLabV3 [1]. Zhang et al. [46] proposed a semi-supervised learning dataset called S5Mars dedicated to Mars images and introduced a semi-supervised learning framework for Martian image semantic segmentation. However, current methods for Martian terrain segmentation generally target processing Martian images from a specific viewpoint and cannot be applied simultaneously to semantic segmentation of terrain from multiple viewpoints, such as those of Mars rovers or satellites. Instead, our proposed MTSNet aims to fully exploits the powerful generalization ability of visual foundation model and the flexibility and efficiency of text prompt-based fine-tuning mechanism, enabling the model to simultaneously model the diversity of Mars terrain from different perspectives within a unified framework, with excellent multi-modal learning capabilities.

### 2.2 Domain-Specific SAM Applications

Recently, the Segment Anything Model (SAM) [19] has attracted widespread attention as a foundation model for prompt-based image segmentation tasks. It can segment images based on flexible and diverse prompts, such as bounding boxes, masks, and points. However, SAM's performance is unsatisfactory for some domains-specific data, such as medical images or images of Mars surface. This can be attributed to the fact that the data used to train SAM is primarily sourced from general images on the earth, and the imaging mechanism or structural distribution of these source data are very different from the target domain data [2, 5, 29]. Nonetheless, some works attempt to break through this difficulty in transfer learning, and have successfully applied SAM to domains-specific segmentation tasks by setting tailored learnable modules and fine-tuning

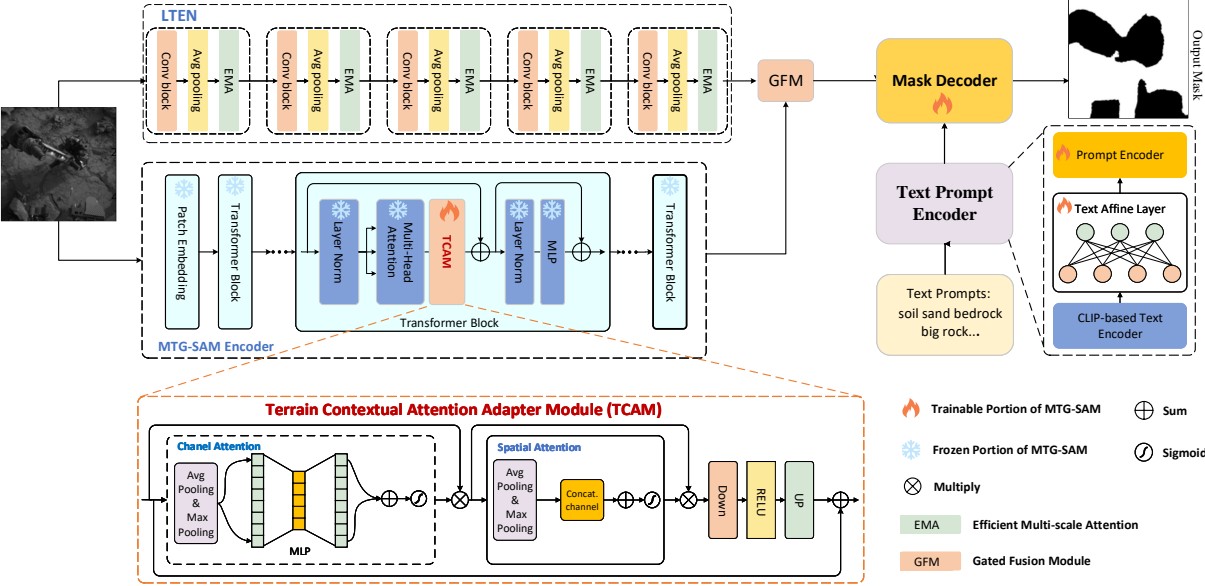

**Figure 1: The overall MTSNet framework: the bottom branch is MTG-SAM encoder equipped with Terrain Contextual attention Adapter Module (TCAM) and the upper branch is a designated Local Terrain Feature Enhancement Network (LTEN, composed of multiple LTE Blocks). These two branches are then integrated through a Gated Fusion Module (GFM), along with a text prompt encoder suitable for text-prompt segmentation. As illustrated, given a Martian terrain image and/or a text prompt describing the terrain of interest as input, MTSNet can outputs the corresponding segmentation mask.**

mechanisms [2, 21, 25]. Ma et al. [25] proposed MedSAM, which only trains the SAM decoder while freezing the encoder parameters. Chen et al. [2] proposed SAM-Adapter, which incorporates domain-specific information and visual prompts into the segmentation network using a simple yet effective adapter technique. Hu et al. [21] proposed AutoSAM that keeps the encoder frozen but adds an independent CNN-based trainable prediction head. However, these methods rely on bounding boxes or foreground points as prompts, requiring costly annotations on images, which limits their application to tasks where data annotation is scarce. Considering the scarcity and re-annotation cost of Mars terrain dataset, we present a novel extended SAM model, referred to as MTG-SAM, that can accept text as prompt input, thus only the terrain text of interest need to be provided, and the model can predict the corresponding terrain segmentation mask, which is crucial for the safe landing and autonomous driving of the Mars rover.

## 3  METHODS

### 3.1  Overview

As shown in Figure 1, our proposed multi-view Martian Terrain Segmentation framework (MTSNet) consist of Martian Terrain text-Guided Segment Anything Model (MTG-SAM) encoder equipped with Terrain Contextual attention Adapter Module (TCAM), a Local Terrain Feature Enhancement Network (LTEN), a Gated Fusion Module (GFM), a text prompt encoder, and a mask decoder by [18]. As illustrated, given an original Martian Terrain image and a text prompt describing the terrain of interest as input, MTSNet can predict the corresponding terrain segmentation mask.

### 3.2  MTG-SAM Encoder with TCAM Module

In original SAM, the image encoder possesses the majority of model parameters, and full model training would incur a high computational cost. Therefore, to integrate Martian terrain knowledge into it at a lower cost, we adopt the Adapter technique [16, 35]. Specifically, during the training process, we froze all parameters of the original image encoder and inserted a lightweight trainable Terrain Context Attention Adapter Module (TCAM) into each Transformer block. The TCAM comprises merely 7.64% of the trainable parameters of the original image encoder and consists of channel-spatial attention to make the image encoder adapt to the domain knowledge of Martian terrains, as shown in Figure 1. In the TCAM, there is first a channel-spatial attention module that learns terrain features from the channel and spatial dimensions.

In the channel dimension, we separately use global max and global average pooling to compress the input feature $X$ with shape $C \times H \times W$ into shape $C \times 1 \times 1$, then pass them through a shared MLP, respectively. The two pooled channel features are then added together, and channel-wise weights are obtained via a sigmoid function and multiplied with the input features. The channel attention process is shown as

$$
\begin{aligned}
Y_c &= \alpha_c \odot X \\
\alpha_c &= \sigma\left(MLP\left(P_{avg}\left(X\right)\right) + MLP\left(P_{max}\left(X\right)\right)\right)
\end{aligned} \tag{1}
$$

where $X$ represents the original input features, $Y_c$ represents the output of channel attention, $\alpha_c$ denotes the channel attention scores, $\sigma$ represents the sigmoid function, $P_{avg}$ and $P_{max}$ denote the global average pooling and global max pooling, respectively.

In the spatial dimension, as shown in Eq. 2, we similarly use two pooling operations to compress the $Y_c$ maps into $1 \times H \times W$, concatenate them along the channel dimension, and recover the channel dimension through convolution. We then use a sigmoid function $\sigma$ to obtain spatial location weights.

$$Y_s = \alpha_s \odot Y_c$$
$$\alpha_s = \sigma \left( f \left[ P_{avg} \left( Y_c \right) ; P_{max} \left( Y_c \right) \right] \right) \tag{2}$$

where $Y_s$ denotes the output of spatial attention, $\alpha_s$ denotes the spatial attention scores, and $f$ denotes the convolution operation.

After the channel-spatial attention module, we further employ convolutional layers to downsample the spatial resolution of the feature maps by a factor of two, and then use transposed convolution to restore the resolution while maintaining the same number of channels as the input, and finally add it to the input feature $X$, which can be described as follows

$$Y = f^T \left( Down(Y_s) \right) + X \tag{3}$$

where $Y$ represents the output of TCAM, $f^T$ represents the transposed convolution.

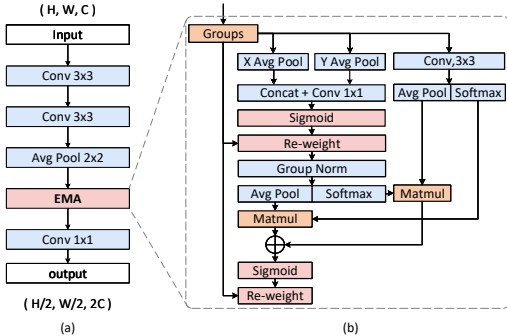

Figure 2: (a) is one of the Local Terrain Enhancement (LTE) Block, and (b) illustrates the detailed Efficient Multi-scale Attention (EMA) module.

### 3.3 Local Terrain Feature Enhancement Network

The Martian surface contains rich geomorphological features, which exhibit unique geometric and textural characteristics. These local features play a crucial role in accurate terrain recognition and contribute to generating more refined segmentation results. However, global context encoder may fail to fully capture these subtle differences. Therefore, we propose a Local Terrain Feature Enhancement Network (LTEN) to enhance the learning of local refinements. Tis composed of the stacks of multiple Local Terrain Enhancement (LTE) blocks. The structure of each LTE block is shown in Figure 2. First, we utilize 3x3 convolutional layers to extract image features while maintaining the image size. After the convolution operation, we use average pooling for downsampling with a scaling factor of 2. Finally, to fuse important terrain features from different spatial locations and scales, we apply an Efficient Multi-scale Attention module (EMA) [26] to improve the model's ability to recognize and distinguish various complex terrain structures.

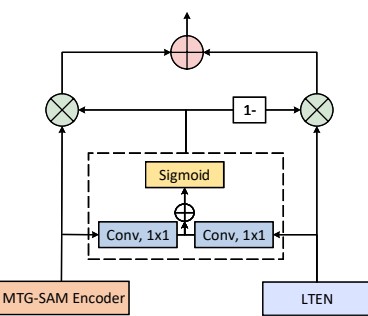

Figure 3: Diagram of the Gated Fusion Module (GFM).

### 3.4 Gated Fusion Module (GFM)

Conventional feature fusion methods such as Concatenation, Addition, and Multiplication directly fuse features without considering the relative importance of the fused features, potentially including redundant information [22]. To address this issue, when fusing local terrain features and global context features, we propose a Gated Fusion Module (GFM) as shown in Figure 3. It can filter out redundant information for each channel and location during fusion process, enabling dynamic and adaptive feature fusion. The formula for GFM is as follows

$$F = (1 - G) \times F_L + G \times F_S$$
$$G = \sigma \left( W_1^T F_L + W_2^T F_S \right) \tag{4}$$

where $F_L$ represents the LTEN features obtained from LTEN, $F_S$ represents the global context features obtained from the MTG-SAM encoder, and $G$ denotes the gating signal. It is obtained by linearly transforming the LTEN features and MTG-SAM global features with weights $W_1$ and $W_2$, respectively, and then applying a sigmoid function.

### 3.5 Text Prompt Encoder

As mentioned above, the current applications of SAM use bounding boxes and points as segmentation prompts. However, in Martian terrain semantic segmentation, the complex structures of Martian terrains make it difficult to effectively distinguish different terrains, and providing prompt annotations requires considerable domain expertise. Not only is this costly, it is also not feasible in practice. We believe text prompts can be utilized for terrain segmentation. To this end, we propose a text prompt encoder on top of the original prompt encoder, enabling the model to perform segmentation using text prompts. As shown in Figure 1, in the text prompt encoder, we first encode the text using text encoder of pretrained CLIP [34] model, which remains frozen during our training process. Since CLIP was not trained on terrain terminology, the generated text embedding have representation limitations. To address this issue, we introduce a trainable text affine layer (TAL) that takes the text embedding from CLIP text encoder as input and transforms them into terrain terminology relevant representations. The relevant operation is as follows

$$y = BatchNorm \left( ReLU \left( W^T E + b \right) \right) \tag{5}$$

where $E$ represents the text embedding from CLIP, $y$ denotes the transformed representations, and $W^T$ and $b$ are the weight and bias of the text affine layer, respectively. By appending the terrain terminology relevant text embedding to the MTG-SAM prompt encoder, the prompt encoder is capable of using text prompts for Martian terrain segmentation

# 4 EXPERIMENTS

## 4.1 Datasets and Evaluation Metrics

We conduct experiments on two Mars datasets to demonstrate the effectiveness and superiority of our proposed method, namely the AI4Mars dataset [40] and the ConeQuest dataset [32].

**The AI4Mars dataset** [40] is the first publicly available large-scale annotated Mars dataset, with images acquired from the Navigation Cameras (NAVCAM) and Mast Cameras (Mastcam) onboard the Curiosity rover. It consists of 16,386 annotated images, among which the training set includes 16,064 annotated images, and the test set contains 322 annotated images. The pixels in each image are classified into four terrain classes: soil, bedrock, sand and big rock. In our study, we focus solely on terrain segmentation. Therefore, we compute the Dice scores of each class and the Intersection over Union (IoU) Metric for comparison, which are defined as follows

$$\begin{cases} Dice\,(G,S) & = \{\,\frac{2*|G\cap S|}{|G|+|S|}, if\,(|G|+|S| \neq 0\,)\,;1, O/W.\} \\ IoU\,(G,S) & = \{\,\frac{|G\cap S|}{|G\cup S|}, if\,(G\cup S \neq 0\,)\,;1, O/W.\} \end{cases} \quad (6)$$

where $G$ represents the ground-truth segmentation mask and $S$ represents the predicted segmentation mask.

**The ConeQuest dataset** [32] is the first publicly available expert-annotated dataset specifically designed for the task of recognizing conical landforms on the Martian surface. This dataset encompasses high-resolution remote sensing images covering three prominent regions on Mars—Isidis Planitia (IP), Acidalia Planitia (AP), and Hypanis (HP)—with over 13,000 samples, each accompanied by detailed metadata. There are two benchmark tasks on ConeQuest dataset: Spatial Generalization and Cone-size Generalization, designated to evaluate a model's performance in unseen regions and on cones of different sizes, respectively [32]. In our study, we follow the evaluation metrics adopted by the Spatial Generalization benchmark task (BM-1) proposed in the ConeQuest dataset[32], including Mask Intersection over Union (Mask IoU or MI), Pixel Intersection over Union (Pixel IoU or PI), Pixel Accuracy (PA), Pixel Precision (PP), and Pixel Recall (PR), to assess the consistency and accuracy of our MTSNet in conical landforms segmentation in the perspective of satellite remote sensing, and we compare our proposed MTSNet with the benchmarked methods.

## 4.2 Implementation Details

In our experiments, we initialize the image encoder with a pre-trained SAM ViT-base model from [18]. We perform data augmentation on each training image, including random rotation up to 10 degrees and random saturation and brightness transformations. For the AI4Mars dataset, we randomly select 2,400 images from the 16,064 training images as the validation set, while the remaining images are used for training. For the ConeQuest dataset, we follow the official split for the training, validation, and test sets. Across all

datasets, we resize the images to 512x512 resolution during training, using a batch size of 8, and train on an RTX 3090 GPU with approximately 14GB of memory required. During training, we employ the AdamW optimizer and fix the initial learning rate at 1e-4.

## 4.3 Experimental Results on AI4Mars

For the AI4Mars dataset, the experimental results are shown in Table 1, where we compute the average Dice scores and IoU scores across the four classes in the test dataset. We compare our method with classical convolutional neural network-based segmentation methods (UNet [36], UNext [43], DeepLabV3 [1]) and Transformer-based methods (TransUNet [9], MedT [41]). Additionally, We also use original SAM with text prompt without any training (called SAM-ZS), and retrain SAM-Adapter [2] and AdaptiveSAM [29] on two datasets for fair comparison.

As shown in Table 1, general semantic segmentation methods perform poorly on Mars terrain segmentation. Among the convolutional neural network-based methods, UNet performs moderately on the "bedrock" and "soil" classes, but excellently on the "big rock" class with a Dice score of 98.45, though its overall performance is mediocre. While UNext shows some improvement on "bedrock" and "soil", it severely fails on the "big rock" class. The performance of DeepLabV3 is between UNet and UNext. As for the Transformer-based methods, MedT and TransUNet adopt Transformer encoders, requiring more data to generate good embeddings. Due to the small scale and class imbalance of the AI4Mars dataset, their performance is unsatisfactory. Moreover, we can see that in the zero-shot case, i.e., SAM-ZS, it is inapplicable to the Mars terrain segmentation task, while SAM-Adapter and AdaptiveSAM, which employ the Adapter based fine-tuning technique, exhibit significant performance improvements, outperforming conventional segmentation methods.Even using only TCAM, our method can achieve the highest scores in most classes compared to other methods, particularly excelling in recognizing "sand" and "big rock". When equipped with LTEN module, the performance is significantly boosted, reaching an average IoU score of 77.37%, increased by 5.71% and an average Dice score of 80.36%, increased by 5.68%, respectively.

Figure 4 illustrates the visual comparison of our method with SAM-Adapter and AdaptiveSAM. It can be observed that the results of our method are closest to the ground truth annotations, with occasional partial prediction errors. In contrast, the results of SAM-Adapter and AdaptiveSAM exhibit over-prediction or under-prediction of terrain features, as well as blurred boundaries in terrain prediction. Using only TCAM in our method, the aforementioned issues are significantly improved. With the introduction of LTEN, it can be seen that the predictions become even more aligned with the ground truth. In addition, MTSNet runs at about 16 frames per second in our experimental environment, which can basically meet the real-time requirements. Although the CNN-based methods have lower computational complexity and faster inference speed, its performance is too weak compared to MTSNet and cannot solve the Martian terrain segmentation task well. Compared with other ViT-based models such as MedT, MTSNet achieves significant advantages in performance and time efficiency. In future work, we plan to explore more efficient methods such as FastSAM [42], EfficientSAM [48], etc. to further improve computational efficiency.

**Table 1: The results on the AI4Mars dataset. Bold: best results, underline: second best results.**

| Method | Dice Score | | | | | IOU Score | | | | |
|---|---|---|---|---|---|---|---|---|---|---|
| | bedrock | soil | sand | big rock | avg | bedrock | soil | sand | big rock | avg |
| UNet [36] | 55.91 | 53.42 | 65.84 | 98.45 | 68.41 | 55.91 | 53.42 | 65.84 | 98.45 | 68.41 |
| Unext [43] | 63.80 | 74.51 | 65.84 | 0.07 | 51.06 | 58.56 | 70.15 | 65.84 | 0.04 | 59.62 |
| DeepLabV3 [1] | 55.03 | 45.90 | 64.91 | **98.45** | 66.07 | 55.03 | 45.31 | 64.91 | **98.45** | 65.98 |
| MedT [41] | 26.18 | 24.49 | 65.84 | 98.44 | 53.74 | 22.12 | 19.13 | 65.84 | 98.44 | 51.39 |
| TransUNet [9] | 58.30 | 68.57 | 65.86 | 0.10 | 49.39 | **63.03** | 64.37 | 65.86 | 0.05 | 47.15 |
| SAM-ZS [19] | 9.73 | 20.00 | 14.33 | 0.19 | 11.06 | 6.86 | 15.89 | 11.69 | 0.19 | 8.66 |
| SAM-Adapter [2] | 59.16 | 68.04 | 78.43 | 93.07 | 74.68 | 54.23 | 63.71 | 75.73 | 92.97 | 71.66 |
| AdaptiveSAM [29] | 62.29 | 67.32 | 73.35 | 93.10 | 74.01 | 57.33 | 63.40 | 70.88 | 93.03 | 71.16 |
| **Ours (TCAM)** | 65.86 | 70.85 | 75.41 | 92.97 | 76.27 | 60.72 | 66.87 | 72.61 | 92.87 | 73.27 |
| **Ours (TCAM + LTEN)** | **67.57** | **78.31** | **82.62** | 92.92 | **80.36** | 62.23 | **74.28** | **80.13** | 92.82 | **77.37** |

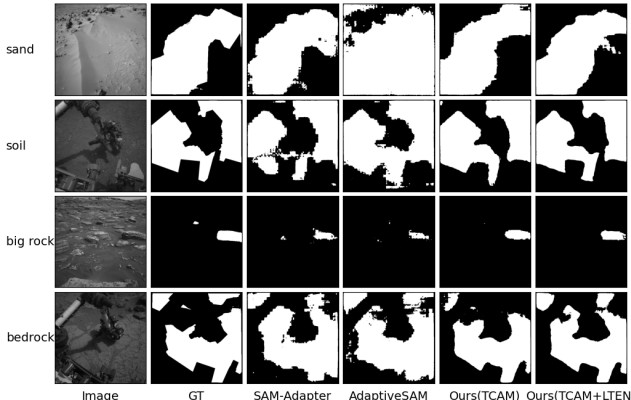

sand
soil
big rock
bedrock

Image    GT    SAM-Adapter    AdaptiveSAM    Ours(TCAM)    Ours(TCAM+LTEN)

**Figure 4: The visualization results on the AI4Mars dataset. From top to bottom, the images of four terrain classes with corresponding text prompts, i.e. "sand", "soil", "big rock", and "bedrock", respectively, as illustrated on the left side of image.**

**Table 2: The results on the ConeQuest dataset in IP region. Bold: best results, underline: second best results.**

| Method | MI | PI | PA | PP | PR |
|---|---|---|---|---|---|
| UNet [36] | 63.46 | 62.33 | 96.27 | 81.92 | 76.44 |
| FPN [23] | **65.94** | 67.91 | 96.62 | 85.96 | 78.00 |
| DeepLabV3 [1] | 34.08 | 31.70 | 93.72 | **97.37** | 32.93 |
| MA-Net [6] | 59.90 | 63.50 | 91.52 | 91.52 | 67.92 |
| AdaptiveSAM [29] | 52.05 | 54.69 | 94.81 | 77.97 | 69.34 |
| SAM-Adapter [2] | 45.96 | 49.32 | 94.87 | 77.35 | 64.42 |
| **Ours (TCAM)** | 60.63 | **69.52** | **97.00** | 87.07 | **79.00** |
| **Ours (TCAM + LTEN)** | 58.88 | 68.44 | 96.86 | 92.52 | 73.20 |

## 4.4 Experimental Results on ConeQuest

On the ConeQuest dataset, we trained on samples from three different regions: Isidis Planitia (IP), Acidalia Planitia (AP), and Hypanis (HP), and separately tested on samples from each region. We compare MTSNet with the official benchmarked models including UNet [36], FPN [23], DeepLabV3 [1], and MA-Net [6], while also including SAM-Adapter and AdaptiveSAM for comprehensive comparison.

For the Isidis Planitia (IP) region, the experimental results are shown in Table 2. UNet, FPN, DeepLabV3, and MA-Net exhibit varying performances, with FPN achieving relatively high levels in Mask IoU and Pixel IoU. AdaptiveSAM and SAM-Adapter perform

relatively moderately, and our method also achieve moderate improvements in Pixel IoU and Pixel Accuracy. We speculate that the mediocre performance of MTSNet is mainly due to 1) the sample and region imbalance of IP region data and 2) the ViT's inherent inability to learn from extremely imbalanced data [49]. The IP region data contains 91% positive samples and only 9% negative samples, which is extremely imbalanced. We believe that proper feature fusion [28], semi-supervised feature learning [39] and tailor-made re-balancing loss functions may alleviate the model's performance degradation on such data.

In the Acidalia Planitia (AP) region, as shown in Table 3, benchmarked models like UNet fluctuates in performance, but FPN maintains a good segmentation accuracy. AdaptiveSAM and SAM-Adapter still perform sub-optimally. On the contrary, our method achieves significant improvements across several metrics, attaining the best scores of 53.21 in Mask IoU, increased by 3.26% and 54.93 in Pixel IoU, increased by 0.88%, respectively. For the Hypanis (HP) region in Table 4, we observe that all methods encounter greater challenges. Compared to other regions, UNet, FPN, DeepLabV3, and MA-Net exhibited declines across various metrics. AdaptiveSAM and SAM-Adapter are more limited in this region. Notably, although our method also faces substantial challenges, it demonstrates adaptability and performance improvement, achieving the best results in Mask IoU, Pixel IoU, Pixel Accuracy, and Pixel Recall, improving Pixel IoU by 2.89% and Pixel Recall by 6.78%.

Figure 5 presents the comparison of the results from our method and other methods on three regions. In the Isidis Planitia region (first row) and the Acidalia Planitia region (second row), for cases where multiple conic landforms exist, other methods exhibit serious missed detection problem, whereas our method ensures no omissions, despite some errors in details. In the Hypanis region (third and fourth rows), compared to other methods, our method does not suffer from recognition errors, while also achieving higher segmentation accuracy.

Through the above experimental results, we can see that benchmarked models like UNet, FPN, DeepLabV3, and MA-Net exhibit varying degrees of performance differences on conical landforms, performing well only in some regions. However, in other regions, when handling unseen distribution features, their generalization capabilities are significantly limited. AdaptiveSAM and SAM-Adapter perform moderately, with metrics noticeably lower than other models. In contrast, our method demonstrates a breakthrough in generalization ability, achieving more outstanding results in the spatial generalization task.

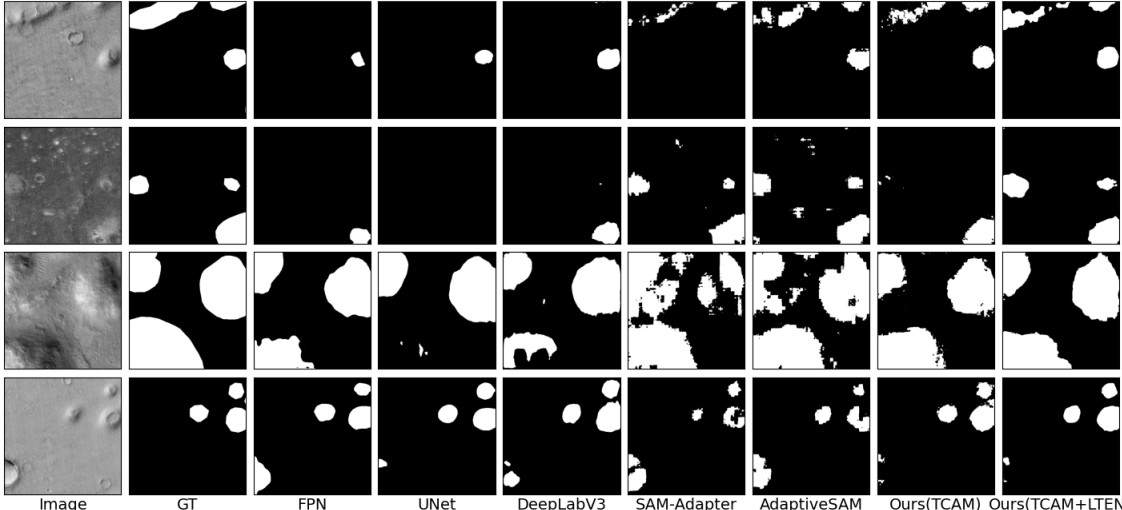

**Figure 5: Qualitative performance of our method and other methods on the ConeQuest dataset.The first row is from region Isidis Planitia (IP), the second row is from region Acidalia Planitia (AP) and the last two rows are from region Hypanis(HP).**

**Table 3: The results on the ConeQuest dataset in AP region. Bold: best results, underline: second best results.**

| Method | MI | PI | PA | PP | PR |
|---|---|---|---|---|---|
| UNet | 47.82 | 42.56 | 95.99 | 82.38 | 47.09 |
| FPN | 48.26 | 45.91 | 96.51 | 87.12 | 47.09 |
| DeepLabV3 | 37.78 | 40.50 | 95.91 | 86.91 | 45.30 |
| MA-Net | 49.94 | 54.05 | **97.20** | 83.96 | 59.85 |
| AdaptiveSAM | 26.72 | 24.08 | 93.18 | 66.04 | 34.55 |
| SAM-Adapter | 21.34 | 20.03 | 93.25 | 69.89 | 28.66 |
| **Ours (TCAM)** | 29.64 | 33.84 | 95.07 | 74.75 | 38.57 |
| **Ours (TCAM + LTEN)** | **53.21** | **54.93** | 97.10 | **87.62** | **63.75** |

**Table 4: The results on the ConeQuest dataset in HP region. Bold: best results, underline: second best results.**

| Method | MI | PI | PA | PP | PR |
|---|---|---|---|---|---|
| UNet | 43.28 | 45.23 | 92.14 | 79.73 | 55.20 |
| FPN | 48.39 | 49.13 | 92.98 | 82.66 | 58.38 |
| DeepLabV3 | 37.93 | 39.83 | 91.63 | 81.11 | 49.48 |
| MA-Net | 42.44 | 44.13 | 92.46 | **82.88** | 51.18 |
| AdaptiveSAM | 37.19 | 38.13 | 80.79 | 70.68 | 53.67 |
| SAM-Adapter | 36.28 | 38.20 | 90.40 | 69.60 | 51.97 |
| **Ours (TCAM)** | 39.41 | 44.38 | 92.45 | 74.04 | 53.51 |
| **Ours (TCAM + LTEN)** | **48.56** | **52.02** | **93.03** | 74.25 | **65.16** |

## 4.5 Ablation Study

To investigate the influence of different model components, we conducted ablation experiments on the proposed model using the AI4Mars dataset, with the following details.

**Effectiveness of the text affine layer (TAL)** : Table 5 shows the performance comparison between the model with TAL and the model without TAL. The results demonstrate that incorporating the TAL as a text domain adapter significantly improves the segmentation performance across various terrain types. Specifically, with the TAL, we observed an average increase of 0.87% in Dice scores and 1.00% in IoU scores. These results validate our approach of

**Table 5: Ablation study of text affine layer (TAL) on the AI4Mars dataset. Bold: best results.**

| Modules | | | Dice Score | | | | | IOU Score | | | | |
|---|---|---|---|---|---|---|---|---|---|---|---|---|
| TCAM | LTEN | TAL | bedrock | soil | sand | big rock | avg | bedrock | soil | sand | big rock | avg |
| ✓ | ✓ | ✗ | 66.60 | 74.49 | 81.90 | 94.96 | 79.49 | 61.63 | 70.36 | 79.12 | 94.86 | 76.37 |
| ✓ | ✓ | ✓ | **67.57** | **78.31** | **82.62** | 92.92 | **80.36** | **62.23** | **74.28** | **80.13** | 92.82 | **77.37** |

using a learnable text affine layer to bridge the gap between CLIP's general text embeddings and the specialized domain knowledge required for Martian terrain segmentation. **Effectiveness of the Gated Fusion Module (GFM)** : When exploring feature fusion strategies, we compared three traditional fusion methods (Addition, Concatenation, Multiplication) with our proposed GFM. The experimental results in Table 6 show that, compared to traditional fusion strategies, GFM achieves the optimal results in feature fusion, especially yielding significant improvements in the "bedrock" and "soil" classes, as well as the overall performance. Specifically, GFM improves the Dice scores and IoU scores by 1.99% and 0.85% over Addition fusion, 1.87% and 2.02% over Concatenation fusion, and 2.29% and 2.19% over Multiplication fusion, respectively. These results demonstrate the clear advantage of GFM in feature fusion. When fusing local features and MTG-SAM global features, GFM comprehensively considers both types of features, selectively filtering information from each channel and position, thereby effectively fusing useful information to improve the accuracy of Mars terrain segmentation.

**Effectiveness of Channel-Spatial Attention in TCAM**: In TCAM, to validate the effectiveness of channel-spatial attention, we compared different attention mechanisms, including Pyramid Split Attention (PSA) [15], Coordinate Attention [14], Efficient Multi-scale Attention (EMA), as well as applying spatial attention and channel attention separately. It should be noted that here we focus on evaluating the impact of different attention mechanisms on the TCAM module, without additionally introducing LTEN. The

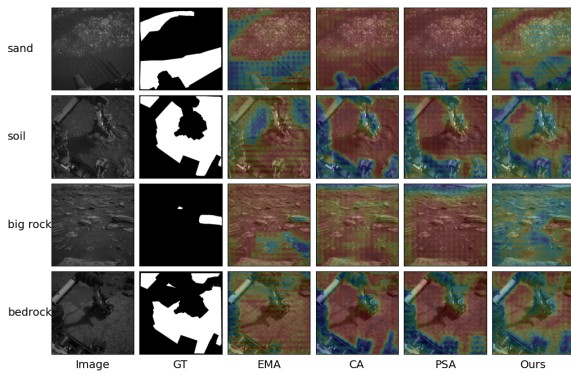

**Figure 6: Visualized comparison of different attention mechanisms in TCAM. EMA: Efficient Multi-scale Attention; CA: Coordinate Attention; PSA: Pyramid Split Attention.**

**Table 6: Ablation experiments with different feature fusion strategies. Bold: best results, underline: second best.**

| Fusion Method | Dice Score | | | | | IOU Score | | | | |
|---|---|---|---|---|---|---|---|---|---|---|
| | bedrock | soil | sand | big rock | avg | bedrock | soil | sand | big rock | avg |
| Addition | 65.58 | 76.69 | 82.83 | 93.02 | 79.53 | 60.26 | 72.53 | 80.19 | 92.92 | 76.47 |
| Concat | 62.35 | 76.50 | 82.17 | 92.99 | 78.50 | 56.83 | 73.37 | 79.32 | 92.90 | 75.35 |
| Multiplication | 65.68 | 76.14 | 81.08 | 90.20 | 78.28 | 60.21 | 71.88 | 78.51 | 90.10 | 75.18 |
| **GFM** | **67.57** | **78.31** | 82.62 | 92.92 | **80.36** | **62.23** | **74.28** | 80.13 | 92.82 | **77.37** |

experimental results in Table 7 show that although various attention mechanisms improve the model's performance to some extent, our channel-spatial attention achieves the optimal results. Specifically, compared to PSA, our method significantly improves the Dice scores and IoU scores across all classes and overall. Compared to Coordinate Attention or EMA, although there are some differences in individual classes, our method has a certain advantage in overall performance. Moreover, applying spatial attention and channel attention separately yields inferior overall performance compared to channel-spatial attention, indicating that handling complex Mars terrain features requires simultaneously considering both spatial position information and cross-channel dependencies.

Figure 6 presents a visualized comparison among different attention mechanisms in TCAM. We generate the heatmap in Figure 6 from the output of the last Transformer block's TCAM. It can be observed that for "sand" and "big rock", the channel-spatial attention more effectively captures the salient features. When dealing with "soil" and "bedrock", the Coordinate Attention, PSA, and channel-spatial attention all exhibit better focusing capabilities. Overall, the channel-spatial attention appears more balanced, demonstrating relatively better performance and stability under various terrain conditions.

**Effectiveness of EMA in LTE Block**: In LTEN, to enhance the learning capability of local terrain features, we incorporated Efficient Multi-scale Attention (EMA) after each LTE Block. To validate the effectiveness of EMA, we designed corresponding experiments, using Pyramid Split Attention (PSA) [15], Coordinate Attention (CA) [14], SE (Squeeze-and-Excitation) [17], and CBAM [47] for comparison. The experimental results in Table 8 show that different attention mechanisms perform variably across terrain classes, but

**Table 7: Ablation study with different attention mechanisms in TCAM. Bold: best results, underline: second best.**

| Attention in TCAM | Dice Score | | | | | IOU Score | | | | |
|---|---|---|---|---|---|---|---|---|---|---|
| | bedrock | soil | sand | big rock | avg | bedrock | soil | sand | big rock | avg |
| PSA [15] | 61.25 | 66.45 | 70.54 | 91.74 | 72.50 | 56.47 | 62.37 | 68.53 | 91.64 | 69.75 |
| Coordinate Attention [14] | 64.08 | **71.28** | 76.84 | 92.72 | 76.23 | 58.74 | **67.41** | 74.01 | 92.73 | 73.20 |
| EMA [26] | 64.06 | 69.77 | **77.78** | 92.59 | 76.05 | 58.95 | 65.99 | **75.19** | 92.51 | 73.16 |
| **Spatial Attention** | 64.35 | 69.08 | 71.18 | 92.71 | 74.43 | 59.23 | 65.45 | 68.60 | 92.62 | 71.48 |
| **Channel Attention** | 64.55 | 67.99 | 73.11 | 92.09 | 74.43 | 59.56 | 64.05 | 70.85 | 91.99 | 71.61 |
| **Channel-Spatial attention** | **65.86** | 70.85 | 75.41 | **92.97** | **76.27** | **60.72** | 66.87 | 72.61 | **92.87** | **73.27** |

**Table 8: Ablation study with different attention mechanisms in LTEN. Bold: best results, underline: second best results.**

| Attention in LTEN | Dice Score | | | | | IOU Score | | | | |
|---|---|---|---|---|---|---|---|---|---|---|
| | bedrock | soil | sand | big rock | avg | bedrock | soil | sand | big rock | avg |
| PSA [15] | 64.91 | 76.16 | 82.28 | 93.92 | 79.32 | 59.49 | 72.19 | 79.77 | **93.82** | 76.32 |
| CA [14] | 65.77 | 77.02 | 82.28 | 93.01 | 79.52 | 60.40 | 72.76 | 79.65 | 92.29 | 76.43 |
| SE [17] | 65.47 | 76.46 | 82.75 | 93.91 | 79.65 | 60.19 | 72.30 | 80.02 | **93.81** | 76.58 |
| CBAM [47] | 66.37 | 76.00 | **82.84** | 92.48 | 79.42 | 60.93 | 71.96 | **80.42** | 92.39 | 76.42 |
| **EMA** | **67.57** | **78.31** | 82.62 | 92.92 | **80.36** | **62.23** | **74.28** | 80.13 | 92.82 | **77.37** |

overall, EMA exhibits the most significant advantage, particularly in the "bedrock" and "soil" classes. Specifically, while PSA excels in recognizing "big rock", and Coordinate Attention, SE, and CBAM perform well in some metrics, but their overall performance is surpassed by EMA. In summary, benefiting from its cross-spatial multi-scale attention mechanism, EMA can better capture local terrain features than other attention methods.

## 5 CONCLUSION

In this paper, we propose a novel text-guided multi-view Martian Terrain Segmentation framework, called MTSNet, which consists of a Martian Terrain text-Guided Segment Anything Model (MTG-SAM) equipped with a lightweight Terrain Contextual Attention Adapter Module (TCAM), a Local Terrain Feature Enhancement Network (LTEN), a simple yet efficient gated fusion module (GFM), a CLIP-based text prompt encoder and a segmentation mask decoder. MTG-SAM is presented to facilitate domain-specific transfer learning from source to target domain data, TCAM is able to fine-tune the MTG-SAM with extremely few adapter parameter, LTEN can well compensate for the limitation of the MTG-SAM encoder in terrain detail capture. GFM enables to dynamically fuse contextual and fine-grained features through an adaptive and learnable gating mechanism. Comprehensive experiments demonstrates that our proposed MTSNet can effectively perform the Mars terrain and cone segmentation with the data acquired from Mars rover and satellite remote sensing, and achieves state-of-the-art performance, providing a more efficient and adaptive solution for Martian terrain segmentation. In the future, we will further lightweight the proposed MTSNet framework and extend it to more general and diverse image segmentation scenarios.

## ACKNOWLEDGMENTS

This work was supported by the National Natural Science Foundation of China under Grants No. 62441601, No. U22A2096, and No. 62221005, in part by the Natural Science Foundation of Chongqing under Grant No. CSTB2022NSCQ-MSX1202.

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
