# OpenReview forum: "MTSNet: Joint Feature Adaptation and Enhancement for Text-Guided Multi-view Martian Terrain Segmentation"
_acmmm.org/ACMMM/2024/Conference — MM2024 Poster_

### Official Review · Reviewer_Qy7c · 2024-05-21

**Rating:** 4
**Confidence:** 3

**Summary:**

This paper presents a novel text-guided multi-view Martian Terrain Segmentation framework, called MSTNet. It consists of an efficient Martian Terrain text-Guided Segmentation Anything Model (MTS-SAM) and a tailored Local Terrain Feature Enhancement Network (LTEN) to capture intricate terrain details. Experimental results on public available datasets show state-of-the-art performance on multi-view terrain segmentation.

**Strengths:**

(1)The proposed MTG-SAM module enables to extract global features,  along with the lightweight TCAM to allow adapting to domain knowledge of Martian terrain segmentation.
(2)The LTEN module focuses on extracting local features with an efficient multi-scale attention module, which is complementary with the above MTG-SAM.
(3)The GFM module integrates the features of MTG-SAM and LTEN well, along with a text prompt encoder for efficient text embedding of Martian terrain segmentation.
(4)The experiments are well organized and state-of-the-art results are achieved compared with most existing methods.

**Limitations:**

(1)The description of the Text Prompt Encoder module is missing in the experiment. Besides facilitating regional retrieval, is it helpful for improving tasks?
(2)The description of the text prompt encoder in Figure 1 and Section 3.5 does not correspond well. The prompt encoder is not described in the text. Where are the learnable parameters?
(3)There is a citation error in the figure, which is to write Figure 2 as Figure 3 on page 4.
(4)There are some grammar errors in the text, including “a Efficient Multi-scale Attention ...” in page 4, “called MTSNet, it consists of ...” in page 8, and etc.

**Suitability:**

2

---

### Official Review · Reviewer_jtcC · 2024-05-24

**Rating:** 4
**Confidence:** 3

**Summary:**

This paper proposed a new framework MTSNet for the Martian terrain segmentation.
It takes in a Mars image and text prompt of interested area to output a segmentation map.
MTSNet combines several existing methods to achieve the goal of Martian terrain segmentation including SAM and Gated Fusion. It designs a new Local Terrain Feature Enhancement Network (LTEN in the paper) to extract and enhance the fine-grained details of the terrains. LTEN consists of convlutional layers and multi-scale attention layers to learn the local information of the terrains. Besides, based on SAM-adapter method, they propose a new module named Terrain Contextual Attention Adapter Module (TCAM) to fine-tune SAM on a specific domain, Martian Terrain domain in this paper. The local information and contexual information is fused by a gated fusion module. Then the fused features are fed into a mask decoder with corresponding text prompt.

**Strengths:**

Compared with the existing methods, MTSNet only uses the text prompt to guide the network, which does not need annotations of bounding boxes. It expolres the possibilites of using text prompt to guide the network and works well on the Martian terrain segmentation. Experiments on the Ai4Mars and ConeQuest dataset show that MTSNet outperforms the existing methods on most of the terrains. Sufficient ablation study shows the necessarity of the proposed LTEN and TCAM Module.

**Limitations:**

1. There is no ablation study of the proposed Text Affine Layer. Also, the author does not explain how and why the Text Affine Layer works. Since it project the CLIP embeddings to the terrain terminology, there is no explanation or experiment to verify the effectiveness of this layer.

The framework is too specific designed for the Martian terrain segmentation. It may not suitable to be used in other domains, which lacks generalization ability.

The source code is not available.

The layout of the four tables could be better.
typo: in line 359, 'LTE block is shown in Figure 2.' not 'Figure 3'

**Suitability:**

2

---

### Official Review · Reviewer_HgUd · 2024-05-24

**Rating:** 3
**Confidence:** 2

**Summary:**

This essay introduces a novel multi-view Martian terrain segmentation framework (MTSNet), designed to meet the needs of autonomous navigation and safe driving of Mars rovers, as well as global analysis of Martian geological landforms for Mars exploration missions. MTSNet combines an efficient Martian terrain text-guided Segment Anything Model (MTG-SAM) with a specially designed Local Terrain Feature Enhancement Network (LTEN) to capture intricate terrain details. The MTG-SAM is equipped with a Terrain Contextual Attention Adapter Module (TCAM) to enhance the model's adaptability and transferability to the specific distribution of Martian terrain. The LTEN is intended to compensate for the limitations of the MTG-SAM encoder in capturing fine-grained local terrain features on the Martian surface. Furthermore, MTSNet introduces a simple yet efficient Gated Fusion Module (GFM) to dynamically merge global contextual features from the MTG-SAM encoder with local refined features from the LTEN module for comprehensive terrain feature learning. MTSNet also supports the use of terrain-specific text as prompts, addressing the issue of costly annotation requirements for bounding boxes or foreground points in existing methods.

The essay also mentions the experimental results of MTSNet on the AI4Mars and ConeQuest datasets, demonstrating that the proposed MTSNet can effectively learn the unique distribution of Martian terrain features and achieve state-of-the-art performance in multi-view terrain segmentation. The introduction of MTSNet brings not only technical innovation but also provides a new efficient tool for terrain analysis in the field of Mars exploration.

**Strengths:**

1.  This work is the first to tackle the multi-view Martian Terrain Segmentation task using only text prompts, achieving superior performance on two benchmarks with notable flexibility and efficiency.
2. The integration of MTG-SAM with LTEN and GFM represents a well-conceived theoretical approach, addressing the complexity of segmenting Martian terrain by considering both global and local features.
3. The paper outlines a technically sound architecture with components like TCAM for adaptability, LTEN for local feature enhancement, and GFM for dynamic feature fusion, along with a text prompt encoder that addresses annotation challenges.
4. The framework is evaluated on relevant datasets (AI4Mars and ConeQuest) using appropriate metrics (Dice scores and IoU scores), and the comparative analysis with existing methods is thorough, demonstrating clear performance improvements and insights from ablation studies.

**Limitations:**

1.The essay has mentioned that the MTSNet might be applied to the self driving of Mars rovers. However, It is well known that autonomous driving requires making correct decisions about the environment in real time, which greatly tests the real-time performance of the algorithm. Unfortunately, the essay does not provide information such as the model's running time, and I believe that this information would be more helpful for an overall evaluation of the work presented in this paper.

2.During the evaluation on the IP region of the ConeQuest dataset, why does the method with TCAM and the LTEN not outperform the TCAM alone, and why does the method only show a modest improvement over the FPN? Does this indicate potential issues with the generalization capabilities of the method? It would be appreciated if you could offer a more in-depth explanation and propose possible plans for enhancement.

3.The essay frequently alludes to the incorporation of a lightweight TCAM component; however, it lacks any comparative data on the model's parameterization. To underscore the benefits of the lightweight design, it would be advantageous to present a comparative analysis of the parameter counts among various attention mechanisms.

**Suitability:**

2

---

### Meta-Review · Area_Chair_kWAY · 2024-07-01

**Recommendation:** Accept (Poster)
**Confidence:** 4

**Metareview:**

In the final ratings, two reviewers lean to accept the paper and one reviewer gives "borderline reject". Prior to the rebuttal, the major concerns are missing analysis on model efficiency, lack of network parameter analysis, and the weak performance on one dataset. In the rebuttal, the authors provide the requested analysis and clear most concerns. The AC agrees with the majority and recommends acceptance.